# Reservoir Conservation in a Micro-Watershed in Tigray, Ethiopian Highlands

**Kazuhisa Koda [1,\*], Gebreyohannes Girmay [2], Tesfay Berihu [2] and Fujio Nagumo [3]**

[1]   Rural Development Division, Japan International Research Center for Agricultural Sciences, 1-1 Ohwashi, Tsukuba, Ibaraki 305-8686, Japan

[2]   College of Dryland Agriculture and Natural Resources, Mekelle University, Mekelle, Tigray, Ethiopia; gebreyohannes.girmay@gmail.com (G.G.); tesra@yahoo.com (T.B.)

[3]   Crop, Livestock and Environmental Division, Japan International Research Center for Agricultural Sciences, 1-1 Ohwashi, Tsukuba, Ibaraki 305-8686, Japan; fnagumo@affrc.go.jp

\*   Correspondence: kodakazu@affrc.go.jp; Tel.: +81-29-838-6676

**Abstract:** Soil erosion in Ethiopian highlands has caused land deterioration due to moving nutrient-rich top soil to downstream reservoirs while leaving reservoirs dysfunctional due to sedimentation. Micro-watershed management by removing reservoir sediments and using them for reclaiming farmland, while using reservoir water for irrigation, can be a potential solution to simultaneously address soil and water constraints and food security challenges. Still, there is knowledge gap before such a solution can be practically applied. The objective of this paper is to present potential solutions for the reservoir sedimentation problem and specifically highlight the utility of bathymetric survey using an echo-sounder to assess sediment volume. Our results indicated that the estimated reservoir sediment volume was 6400 m$^3$ leading to a reclamation of 3.2 hectares by layering 0.2 m sediment. The sediment used for reclamation depicts neutral pH (7.3), high organic carbon (2.5%), available phosphorus (9.2 mg/kg) and exchangeable potassium (25 cmol(+)/kg). Garlic (*Allium sativum*) was planted in the reclaiming abandoned farmland and produced 7.1 t/ha of bulb on average. There is a potential of producing 2–3 horticultural crops per year. Thus, developing methods for scaling up potential farmland reclamation using reservoir sediment would contribute to degraded farmland restoration and food security in Ethiopia and beyond.

**Keywords:** bathymetric survey; reservoir sediment; farmland reclamation; micro-watershed; Ethiopia

## 1. Introduction

The Federal Democratic Republic of Ethiopia (hereinafter Ethiopia) is located in the eastern part of sub-Saharan Africa (latitude: 3–14°N, longitude: 33–48°E). Its landscape is divided by the Great Rift Valley of Africa, a continuous trench that runs through the country in a northeastern-southwestern direction and characterized by diverse agro-ecological systems that greatly vary with topographic elevation, ranging from 125 m below sea level to 4620 m above sea level. During the rainy season, rainfall runs down trench slopes into rivers to form a deep V-shaped valley. Ethiopia, with a population of 104.96 million that annually increases by 2.5%, depends heavily on its agro-ecologies and natural resources for its economy, livelihoods and food security [1]. The agricultural sector, which is dominated by smallholder farmers, accounts for 34% of the GDP, 68% of employment, and a dominant land use system of the country [1].

Ethiopian highlands in the western part of the Great Rift Valley feature a tropical climate, while those located around the perimeter of the Valley can be classified as semi-arid areas. Rain usually starts in mid-June and ends in mid-September. The lack of freshwater resources to meet the water demand

is a critical problem of agriculture in Ethiopian highlands. Irrigated land accounts for about 0.5% of total agricultural land. Rapid population growth, dry climate, and poor availability of arable land place agricultural land under high stress and drive land degradation and desertification. For example, the Tigray region is a typical Ethiopian highland, characterized by chronic droughts due to irregular rains, which cause frequent harvesting failures. Its landscapes with steep slopes are prone to land degradation which is aggravated by conventional tillage practices. While inhabitants in the highlands of Tigray grow wheat as a staple food, the wheat sowing to tillering period, during which uncovered bare soil is exposed to erosion, coincides with the rainy season for about three months, resulting in massive soil erosion. Subsequent soil nutrient loss through surface runoff is responsible for low crop productivity and chronic food insecurity in the region.

A multitude of counter-measures have been taken to address water constraints and soil erosion on the Ethiopian highlands, including efforts to construct reservoirs for irrigation as well as for the protection of watersheds [2–5]. For example, reservoir construction in Tigray started in late 1970s, and 92 reservoirs were built between 1992 and 2012 [6]. In turn, soil conditions in northern Ethiopian highlands, characterized by sedimentary, igneous, and metamorphic rocks [6], combined with irregular rainfall on steep landscapes, cannot completely prevent erosion. To date, soil erosion in the form of sheet, rill, and gully erosion in the watersheds has led to significant deterioration of agricultural land due to exporting top soil from farmland to downstream reservoirs, while leaving reservoirs unfunctional due to soil sedimentation, water leakage, insufficient inflow, structural degradation, spillway erosion, etc. [6].

Ecologically and hydrologically related problems of reservoirs in the region have been extensively investigated [3,4,7–10] regarding the occurrence of cyanobacteria, annual sediment yield of about 19 t/ha/year, and irrigation capacity reduction up to 33% due to water loss through seepage and evaporation, etc. Reservoir sediments were found to consist of fine clay rich in micro-nutrients essential for crop growth [11,12]. Therefore, Girmay et al. [11,12] proposed the management of the micro-watershed by periodically removing nutrient-rich reservoir sediments and using them for reclaiming farmland, while using reservoir water for irrigation as a potential solution to simultaneously address soil and water constraints as well as food security challenges in Tigray and beyond. Still, there is a significant information and knowledge gap to apply and scale-up such a solution in practice. In particular, there has been little research done so far to assess the volume of reservoir sediments, except a few studies by Haregeweyn et al. [2], which found that it is essential to estimate farmland areas to be reclaimed along with the volume of irrigation water available, and to carry out farmland reclamation as well.

The objective of this paper is to present potential solutions to solve the reservoir sedimentation problem in Tigray by presenting a case study of the Adizaboy reservoir in Tigray and documenting the engineering process of implementing farmland reclamation. This study specifically highlights the utility of employing the bathymetric survey method to assess the volume of reservoir sediments, which can be used to estimate the bottom surface even in cases where design reports of reservoirs were not available and further to estimate theoretical areas of farmland to be reclaimed in Tigray and beyond.

## 2. Materials and Methods

### 2.1. Study Site

The Adizaboy micro-watershed (Figure 1) is located at 13.64 to 13.68° N and 39.56 to 39.6° E, about 20 km north of Mekelle city (Kilte Awulaelo District, Tigray region) and spans altitudes of 2050–2275 m above sea level. The watershed features a catchment area of about 8.5 km$^2$, with the average slope of about 8%. The annual rainfall in Wukro near this area varies between 300 mm and 1000 mm, most of which (about 74%) occurrs in July and August, while the average maximum and minimum monthly temperatures equal 30 °C and 8 °C, respectively.

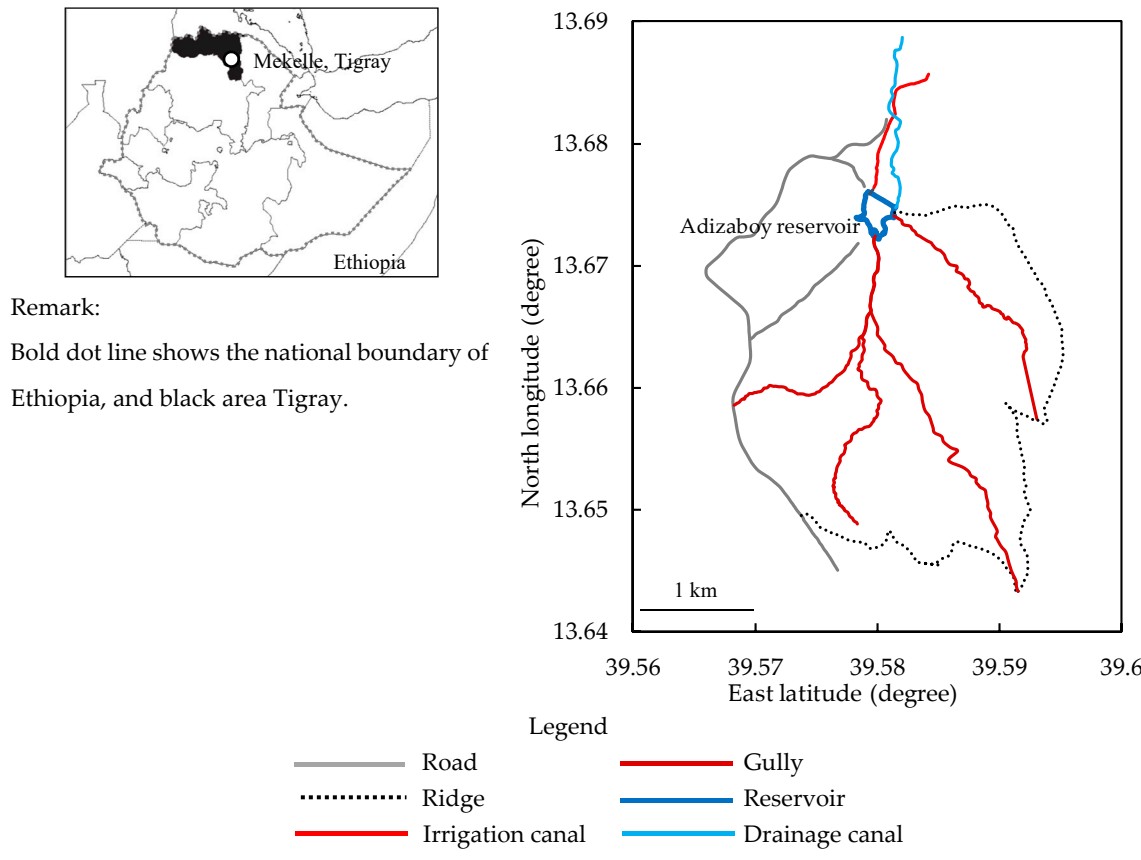

Remark:

Bold dot line shows the national boundary of

Ethiopia, and black area Tigray.

**Figure 1.** Overview of the Adizaboy micro-watershed in northern Ethiopia.

The Adizaboy reservoir is located at the outlet of the Adizaboy watershed, and midstream of the Agulae river watershed. The Adizaboy reservoir was constructed in 2009 by making use of on-site soils and rocks excavated near the construction site, and the underlying geological materials are dominantly weathered shale, marl, and limestone [8]. The Adizaboy reservoir is classified as a zoned fill dam and features a crest length of 276 m, a free board of 2.05 m, and a highest water depth from the bottom pipe of 7.01 m. According to Berhane et al. [6], the Adizaboy reservoir dimensions are smaller than the average of the 92 reservoirs in Tigray.

Some gully erosion formed upstream of the watershed have caused sedimentation of the Adizaboy reservoir. The reservoir sometimes dries out in March and April before the end of the dry season, and the revealed surface soil has been shown to mainly comprise Cambisols and thus be acceptable for agricultural use [13].

The downstream upland areas of the Adizaboy reservoir are used for the cultivation of cereals such as wheat, teff, and barley. After the harvest of these cereals, grass and chick peas are commonly planted with residual moisture and some supplemental irrigation, which prevails to increase the organic carbon and nitrogen contents of soil and save fertilizers. After the rainy season, vegetables such as tomato, onion, and garlic are cultivated in some upland fields.

The Adizaboy dam reservoir shows seasonal changes in its water body cover (Figure 2). Farmland to be reclaimed with the reservoir's sediment is located close to the Adizaboy dam which has water management facilities such as the bottom pipe on the left bank side, spillway on the right bank side, and a rock-covered embankment between them. It has an upstream slope of H3:V1, and a downstream slope of H2.55:V1. However, the above-mentioned problems of sedimentation and leakage, etc. have been observed at the Adizaboy dam reservoir.

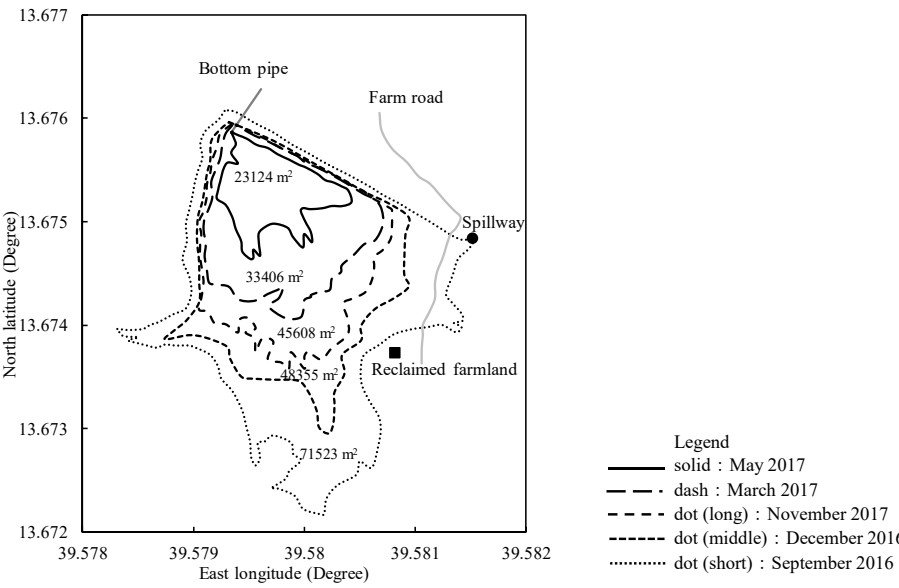

**Figure 2.** Seasonal change of the Adizaboy water body.

## 2.2. Storage Water and Sediment Volume Determination

The storage curve was established using the observation results obtained through a series of bathymetric surveys implemented during September 2016, December 2016, March 2017, May 2017, August 2017, and November 2017 in the Adizaboy reservoir. The underwater depth and features were mapped using a Bosch GOL 26D Professional (Robert Bosch GmbH, Gerlingen, Germany) for level survey, and a GPSMAP 60CSx (Garmin International Inc., Yokohama, Japan) for recording the coordinates along the perimeter of the reservoir surface area (Figure 2).

An echo-sounder HE-8301F-Di (Hondex Inc., Saitama, Japan) and a GPS tracker GPSMAP 60CSx were used to measure water depth and coordinates, respectively, assisted by a small inflatable rubber boat. The bottom sediment surface was calculated by subtracting the water depth obtained using the echo-sounder from the water surface. The reservoir storage volume was analyzed using Surfer Ver. 15 (Golden Software Inc., Colorado, USA) software, which was also used to interpolate known values and draw contour maps of the reservoir bottom. Sediment volume in the Adizaboy reservoir was estimated from the difference between the reservoir bottom depth determined with echo-sounder and the deposited sediment depth determined without echo-sounder.

## 2.3. Water Balance Estimation

The water balance, determined by water inflow and outflow of reservoirs and surrounding components, can be expressed as Equation (1):

$$\frac{dS}{dt} = (P - E) \times A + (R_i + G_i) - (R_0 + G_0) \tag{1}$$

where $S$ is the reservoir storage volume (m$^3$), $P$ is precipitation (m), $E$ is the water volume evaporated per unit reservoir surface (m), $R_i$ and $R_0$ are the surface inflow and outflow, respectively (m$^3$), $G_i$ and $G_0$ are the groundwater inflow and outflow, respectively (m$^3$), $A$ is reservoir area (m$^2$), and $t$ is time (year). The period corresponding to zero storage volume change usually equals one year, and unknown values are generally obtained by substituting known values as residue.

On-site weather observations commenced upstream of the Adizaboy micro-watershed on 21 May 2017. Climatic data such as temperature, humidity, evaporation, and precipitation etc. were collected at 15-minute intervals and downloaded to a notebook PC (Panasonic Corporation, Tokyo, Japan) with pre-installed HOBO software (Onset Computer Corporation, Massachusetts, USA). The weather observation device (Onset Computer Corporation) was assembled, fixed to a metal

stake, and surrounded with a metal fence (perimeter = 16 m, height = 1.5 m) in the backyard of a selected resident's house. Rainfall was automatically observed using a tipping bucket rainfall gauge in this weather observation device. However, annual average precipitation was calculated by using long-term precipitation data collected for the last 22 years in Wukro, located about 20 km north of the Adizaboy micro-watershed.

A data logger (DIK-603C Baro diver, Daiki Rika Kogyo Co., Ltd., Saitama, Japan) was attached to the weather observation device to record atmospheric pressure, and another data logger (DIK-615A diver, Daiki Rika Kogyo Co., Ltd.) was installed in the simple evaporation measurement device (UIZ-PE100, Uizin Co., Ltd., Tokyo, Japan) to observe the accumulated evaporation volume. A white foamed polystyrene cover providing sun protection was used to avoid temperature increases in the evaporation measurement device, which was buried in the ground.

The micro-watershed is a basic unit of the water cycle. Rainfall in the Adizaboy micro-watershed resulted in an influx of surface run-off and groundwater into the Adizaboy reservoir and hence increased the water depth therein. Area ($A$) was obtained from the bathymetric survey. Surface run-off (i.e., $R_i$) and groundwater (i.e., $G_i$) parameters were calculated from the storage curve. While $R_0$ could rarely be observed when there is overflow from the dam's spillway, we simply assumed the annual rainfall that does not cause overflow through the spillway which could happen with floods occurring about once in 10 years. $G_0$ was calculated from Equation (1) above.

Water pressure at the reservoir bottom, which was used for water depth observation, was measured by a data logger (DIK-615A diver). The data logger was attached to the cap of an all-strainer 4-m-long PVC (Polyvinyl Chloride) pipe using a wire, and the PVC pipe, supported by a three-legged metal stand, was placed on the flat bottom surface near the bottom pipe inlet in the Adizaboy reservoir. Air pressure was recorded by another data logger (DIK-615A Baro diver). The two data loggers were activated with a 15-min interval, and water depth was calculated by subtracting air pressure from water pressure, which allowed the hydrograph to be drawn.

Supplemental irrigation through the bottom pipe of the Adizaboy reservoir started in late September, when the rainy season was over, and finished in October. The daily irrigation time was estimated as 10 hours. Flow velocity was measured by a float and a stopwatch in a 4-m section of a straight, steady, and constant-dimension concrete canal (width = 33 cm, height = 45 cm), and the average value ($n$ = 5) was multiplied by 0.85 [14]. One-day livestock water [15] supply for 200 cows, 150 sheep and 150 goats, multiplied by two for one day, was considered.

## 2.4. Farmland Reclamation

The farmland reclamation was conducted in 2017 before the start of the rainy season (mid-June) [16]. The Adizaboy reservoir sediment was excavated and transported to the farmland to be reclaimed (Figure 2) by donkey using stepping stones on the soft wet sediments in the reservoir. To prevent farmland leakage, a stone bund was constructed over approximately two weeks in three stages. In the first stage, locally available stone materials were arranged along the planned ground perimeter, while the second stage featured sediment transport and farmland leveling. In the third stage, the ground around the farmland to be reclaimed was excavated to establish a temporary waterway to divert runoff. In this way, farmland for reclamation was constructed on a gentle slope of bare land around the Adizaboy reservoir.

The farmland to be reclaimed with dimensions of 23 m × 14 m (area = 322 m$^2$) was surrounded by barbed wire and metal poles (perimeter length = 140 m, height = 1.8 m) for security reasons, and the corresponding construction work took two weeks. Garlic (*Allium sativum*) cultivation was started in August and was finished in December 2017. As the sediment depth on the farmland to be reclaimed equaled 0.2 m, shallow-rooted garlic, which does not need deep soil, was selected for culturing. Basin irrigation was conducted using water taken from the Adizaboy reservoir. Conventional row planting of garlic with five replications was carried out with a spacing of 0.2 m between plants on the reclaiming farmland where supplemental irrigation water taken from the Adizaboy reservoir was

delivered once a week. The amount of irrigation water was about 20 mm/day. Fertilizer was not used. Soil samples from bare land around the reservoir and sediment samples were taken at the center and near the perimeter of the Adizaboy reservoir area. Collected samples were analyzed using standard laboratory procedures.

## 3. Results

### 3.1. Reservoir Storage and Sediment Volume

The survey results indicated that the water depth of the Adizaboy reservoir can reach a maximum of about 7 m (Figure 3). The water storage volume as calculated from storage curve developed for this reservoir was estimated at 200,000 m$^3$ ($R^2$ = 0.99) when the dam is full.

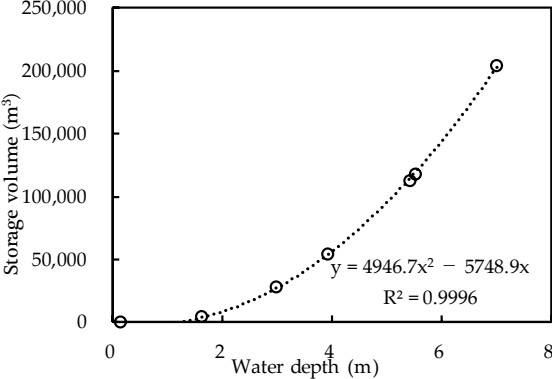

**Figure 3.** The storage curve of the Adizaboy reservoir.

The contour of the Adizaboy reservoir bottom was mapped (Figure 4). Its water surface area was constructed by taking observation points using GPS (Figure 4a), its water depth using level survey without echo-sounder (Figure 4b) and with echo-sounder (Figure 4b). This shows the reservoir bottom estimated by water depths considering the reservoir sediments. The difference between Figure 4b,c indicates the estimated reservoir sediment volume to be 6400 m$^3$. This can be translated to about 3.2 ha of reclaiming farmland area could be cultivated when a 0.2 m layer of sediment is put on the reclaiming farmland.

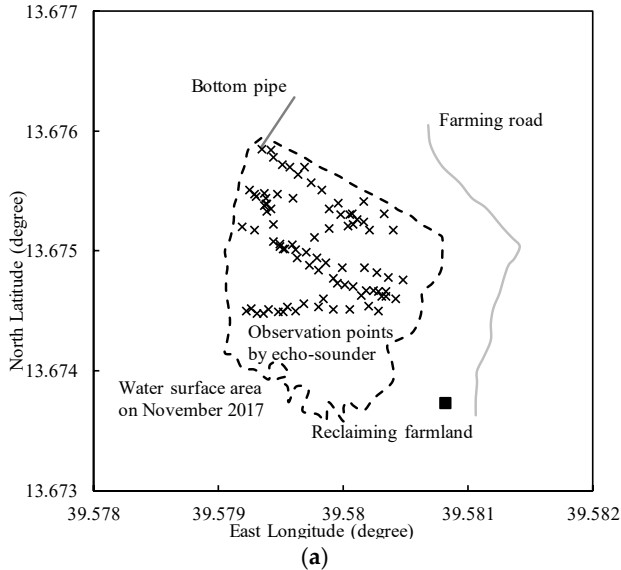

**Figure 4.** *Cont.*

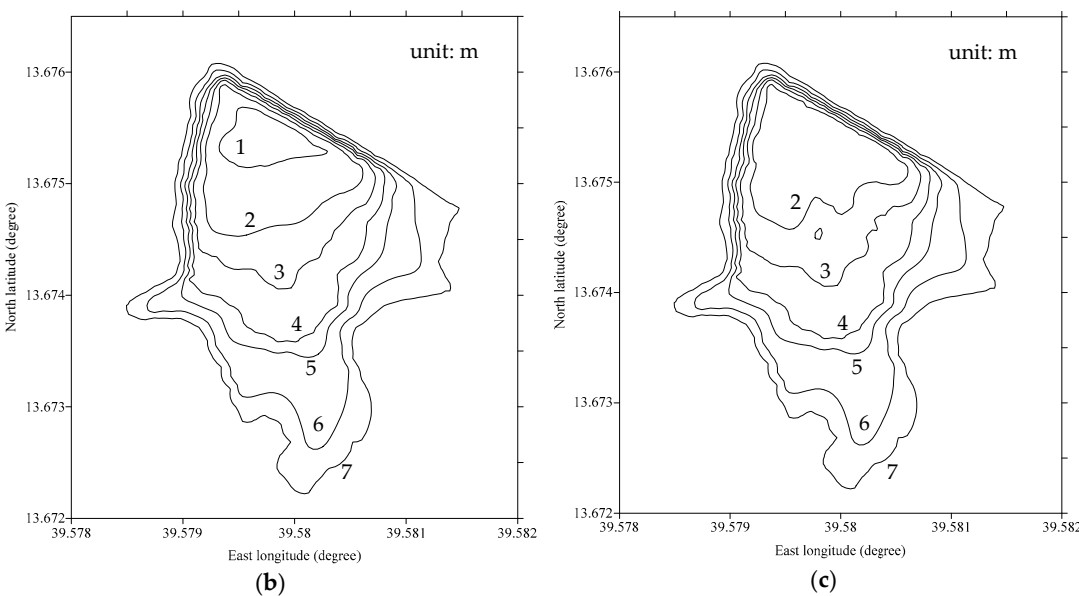

**Figure 4.** The reservoir sediment survey of the Adizaboy reservoir: (**a**) The observation points (X mark) and water surface area; (**b**) The reservoir bottom contour map without echo-sounder; and (**c**) The reservoir bottom contour map with echo-sounder.

The sediment in the Adizaboy reservoir is characterized by fine sands found at the center of the reservoir, and coarse sands near the perimeter of the reservoir (Table 1). Reservoir sediment was classified as cohesive soil that could strongly retain the nutrients of exchangeable potassium in fertilizers and water, but exhibited poor air permeability in comparison to bare soil. The sediment has a neutral reaction (pH 7.3 vs. pH 8.1 on bare land) and contains a high percentage of organic carbon (2.5% vs. 1.8% on bare land), available phosphorus (9.2 mg/kg vs. 8.8 mg/kg on bare land) and exchangeable potassium (25 cmol(+)/kg vs. 14.1 cmol(+)/kg (9.8 g/kg vs. 5.5 g/kg) on bare land).

**Table 1.** Bare land soil and reservoir sediment sample analyses of the Adizaboy micro-watershed.

| Items | Bare Land | Sediments |
|---|---|---|
| Sand (g/kg) | 102 | 70 |
| Silt (g/kg) | 501 | 536 |
| Clay (g/kg) | 397 | 394 |
| Bulk Density (g/cm$^3$) | 1.2 | 1.1 |
| Field Capacity (*v/v*%) | 15 | 28.8 |
| Permanent Wilting Point (*v/v*%) | 9.2 | 13.3 |
| Available Water Capacity (*v/v*%) | 5.9 | 15.6 |
| Available Water Capacity (mm/15cm depth) | 8.8 | 23.3 |
| pH (H$_2$O) | 8.1 | 7.3 |
| Organic Carbon (g/kg) | 17.6 | 24.7 |
| Total Nitrogen (g/kg) | 3.4 | 3.3 |
| Available Phosphorus (mg/kg) | 8.8 | 9.2 |
| Exchangeable Potassium (cmol(+)/kg) | 14.1 | 25.0 |

### 3.2. Reservoir Water Balance

The hyetograph/hydrograph (Figure 5) for the reservoir in the Adizaboy micro-watershed was obtained from rainfall data obtained from the weather observation device installed upstream of the micro-watershed and reservoir storage data. The run-off efficiency of the Adizaboy micro-watershed was estimated at 0.31 based on the hyetograph/hydrograph. The delay time of peak inflow ($G_i$ and $R_i \approx 5149$ m$^3$) from watershed to the reservoir after a selected dry spell in August 2017 was estimated at 17 hours as the watershed is dry and stony.

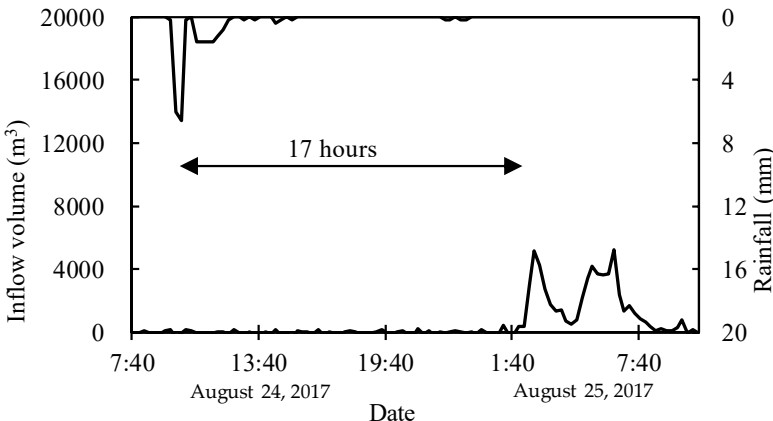

**Figure 5.** The hyetograph/hydrograph of the reservoir in the Adizaboy micro-watershed.

The water balance of the Adizaboy reservoir (Figure 6) was calculated based on long-term annual average precipitation of 600 mm (1992–2013) taken from the Wukro weather station (about 20 km north of the study area) in addition to our own weather observation. The rainfall from the micro-watershed was estimated to be 1,567,700 m$^3$, while rainfall on the reservoir surface was estimated to be 42,914 m$^3$. The evaporation volume (118,800 m$^3$) determined between November 16, 2017 and November 21, 2017 (6.5 mm/day) was almost consistent with the value of 1800 mm/year obtained at the Ziway lake (elevation = 1636 m, area = 440 km$^2$), which is located about 160 km south of Addis Ababa [17].

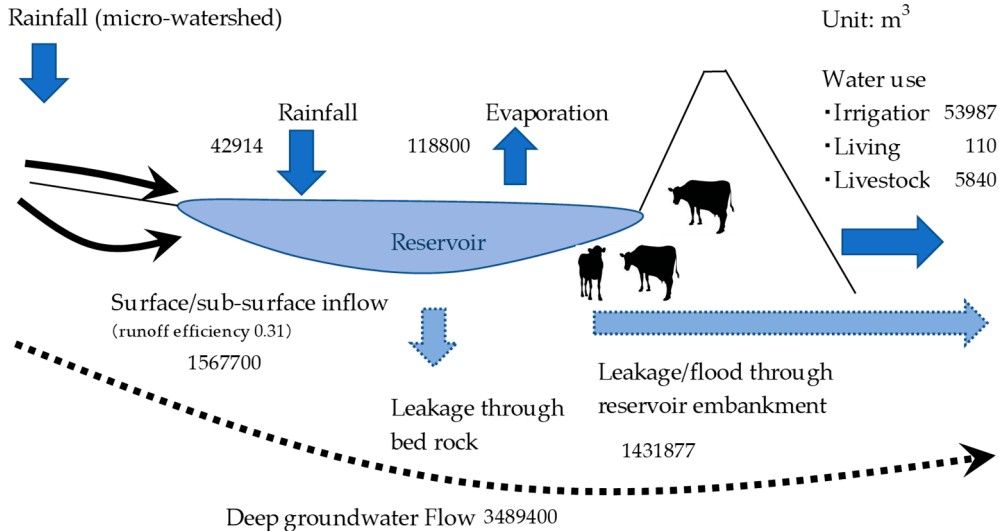

**Figure 6.** The water balance of the Adizaboy reservoir.

The surface and sub-surface water flow to the reservoir was found to have an efficiency of 0.31 (1,567,700 m$^3$), and the remining was estimated to be lost in deep groundwater flow (3,489,400 m$^3$) under the reservoir. Deep groundwater flow, which did not affect the reservoir water balance, was estimated to exceed reservoir inflow (1,610,614 m$^3$) almost two-fold. The water consumption estimated at 59,937 m$^3$ was used for irrigation (53,987 m$^3$), living (110 m$^3$) and livestock drinking (5840 m$^3$).

Average detention time, defined as the time to replace the water stored in the reservoir under steady-state conditions, was estimated as 1.1 years by dividing storage volume by the volume of the outflow (e.g., irrigation water, daily life water, and livestock water). In fact, the Adizaboy reservoir sometimes dried up in March and April and started refilling in May, i.e., during the short rainy season. Compared to the average detention time of lakes in Japan (17 years) [18], the above value was much

smaller, which was ascribed to the fact that the investigated reservoir had a volume much smaller than that of a typical lake.

Considering a long-term annual average rainfall of 600 mm, the potential irrigable area of the Adizaboy reservoir was estimated to be about 27 ha (from a linear irrigable area–rainfall relationship with $R^2$ = 0.99, Figure 7). The volume of groundwater leakage and storage water components in the water balance equation can be used as potential irrigation water. The potential irrigable area is also dependent on efficient irrigation methods (such as the drip method) and irrigation season. The irrigation water depth for a drip irrigation (20 mm/day) system during a dry season (270 days) is about 5400 mm, which could suffice to irrigate the reclaiming farmlands using reservoir sediment leading towards reservoir conservation.

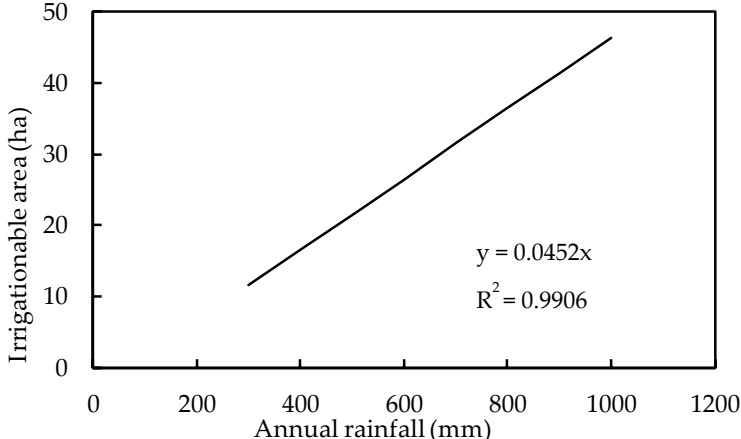

**Figure 7.** The irrigable area of the Adizaboy reservoir.

### 3.3. Farmland Reclamation

In scoping for reservoir conservation through the excavation of deposited sediment in the Adizaboy reservoir (6400 m$^3$), about 3.2 ha of bare land can be reclaimed into farmlands by layering sediment. In practice, garlic was planted on reclaiming farmland in 2017 and 2018. The production of garlic bulb yield ranged from 5.67 t/ha to 8.64 t/ha in 2017 (Figure 8), and from 6.14 t/ha to 8.74 t/ha in 2018 (with the mean yield being 7.1 t/ha). Considering irrigated crops, there is the potential of producing 2–3 horticultural crops per year in the study area once the land reclamation process is complete.

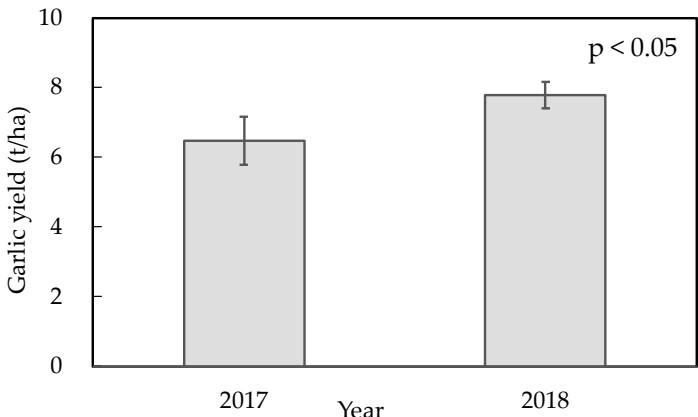

**Figure 8.** Garlic yields on reclaiming farmland by layering sediment collected from the Adizaboy reservoir.

## 4. Discussion

This study presented potential solutions to solve the reservoir sedimentation problem in Tigray by documenting a case study to conserve the Adizaboy reservoir along with the implementation of surveys on reservoir sediments, storage volume, and water balance of the reservoir. The farmland to be reclaimed by using the reservoir sediments was demonstrated as a potential solution for mitigating reservoir sediment deposition. These results support that sediments can be used for vegetable cultivation as they have higher fertility than bare land, while it is necessary to overcome water leakage from the reservoir to make an effective use of the storage water for irrigation.

Methods of soil and water conservation in Ethiopia can be classified into biological (farming soil conservation) and physical (civil engineering) methods, and the combination of these techniques is expected to efficiently prevent soil erosion in the micro-watershed. This method of farmland reclamation by using the sediments in the reservoir presented would be the hybrid method and could be applicable to other reservoirs similarly suffering from sedimentation problems. Girmay et al. [12] conducted garlic cultivation at reclaiming farmland by using sediments from the Maileba reservoir and showed that layering sediments on bare land at a thickness of 0.15 m was better than that of 0.3 m in terms of the cost–benefit ratio. The sediments in the reservoir contain many plant-available nutrients [11] that could be used as nutrient/fertilizer sources for not only vegetable cultivation but also forest, wheat, barley, teff etc. production.

We described the manual excavation of reservoir sediment and watershed management as ways of mitigating water-induced erosion. The former technique involves sediment excavation and transport to farmland to be reclaimed and requires the use of donkeys and local labor force, while the latter option mitigates water erosion by conducting conservation agriculture such as the one being tested on wheat farmland (data not shown) upstream of the Adizaboy micro-watershed. Machine excavation and lost storage replacement are expensive options for farmers to perform on their own. Sediment flushing is necessary if reservoir sedimentation proceeds to levels higher than the bottom pipe inlet.

Run-off and sediments from regions upstream of the watershed move in gullies to the reservoir located at the outlet. Sediment storage dams, comprising stones and gabion, can be used for sediment retention at micro-watershed outlets [9]. Although the use of sediment storage dams is an effective measure supplying short-term benefits, these dams are filled with sediment over several years. Herein, farmland reclamation was conducted on bare land experiencing severe soil erosion, since the top soil layer was excavated to construct the Adizaboy dam. Consequently, no crop cultivation could be performed on the area striped of its top soil as fill material for the dam building. For this reason, the reservoir sediment was layered on bare land to reclaim it and produce crops.

We have to acknowledge there are some limitations in this study. (1) The sedimentation has already been in progress since the Adizaboy reservoir was constructed. In terms of sediment change, reservoirs in general can be classified into three types, namely those with constant, increasing half-way, and reducing half-way sedimentation rates [19]. If the Adizaboy reservoir is of a constant-sedimentation-rate type, which more than half of reservoirs Japan are, it will need a lot of time to be filled up by sediments. In the case where the reservoir sediment survey was conducted soon after the rainy season was over, the sediment volume might have been slightly increased due to the higher water level. There has been no report so far that the flood caused by the reservoir sedimentation destroyed the embankment in Tigray. (2) More than 80% of inflow goes out of the Adizaboy reservoir in leakage. The Adizaboy reservoir is located both at the outlet of a micro-watershed and midstream of a larger watershed featuring the Agulae river as the outlet. Thus, in the Adizaboy micro-watershed, the amount of surface run-off is smaller than that of groundwater, and the Adizaboy reservoir can thus be classified as a flow-through type according to Born et al. [20]. In terms of geological structure, the overlying limestone layer was inclined downward in a South–North direction by about 2.5% [21]. The effective use of ground flow as irrigation water in semi-arid areas is of high importance. Available irrigation water could be estimated and quantified by calculating the reservoir water balance. However, in order for this inflow to be efficient, geological strata should be inclined to

the reservoir. (3) When new vegetable cultivation starts on the reclaiming farmlands, landless farmers will need some incentives. Farmers allotted to reclaiming farmland around the Adizaboy reservoir live in Agulae village located several kilometers downstream and access their land via a highway and farming roads. Some of these farmers plan to construct temporary housing to stay near the reclaiming farmland during the busy farming season. Therefore, we plan to provide such incentives, e.g., planting cash and fodder crops such as aloe vera and cactus, etc. Importantly, we started garlic cultivation with basin irrigation on reclaiming farmland in 2017/2018, and the yield in 2018 was significantly increased. This might be caused by a drought that occurred in 2017 because the garlic was planted in August during the rainy season. (4) Water-saving irrigation such as drip irrigation and use of water harvesting tanks should be realized. Additionally, even though the local government publicly recommends the usage of irrigation water for vegetable cultivation, many farmers use this water for growing wheat, teff, and barley, which leaves plenty of room for farming practice improvement. Further investigation is necessary. (5) Salt damage might cause productivity reduction on reclaiming farmland if the salt accumulation occurs. As far as EC (Electric Conductivity) values in the reservoir water are concerned, it would not occur. However, we should conduct ion chromatography analysis if we use the fertilizer to cultivate the vegetables on reclaiming farmland in the future.

## 5. Conclusions

It is necessary to apply water harvesting techniques and to construct reservoirs to make efficient use of water resources in Tigray, Ethiopia. Although many reservoirs have been constructed, numerous reservoirs located at the outlet of the micro-watershed experience the problems of sediment accumulation [2,22] and water leakage [8]. Herein, on-site survey of the present state of the Adizaboy reservoir suggests undertaking counter measures against sediment accumulation, showing that the effective storage volume of the reservoir can be increased and sustained by: (1) using reservoir sediments for farmland reclamation, and (2) utilizing storage water for the irrigation of vegetable crops.

The current storage volume of the Adizaboy reservoir could be estimated by bathymetric survey. Substantial sediment volume has accumulated in the Adizaboy reservoir. To solve this problem, we have removed part of the deposited sediment and used it to reclaim farmland around the reservoir. The results of the bathymetric survey using an echo-sounder indicated that the estimated reservoir sediment volume was 6400 $m^3$, leading to a reclamation of 3.2 hectares by layering 0.2 m of sediment. Garlic was produced at 7.1 t/ha of bulb on average on reclaiming farmland. There is a potential of producing 2–3 horticultural crops per year.

The vegetable cultivation on reclaiming farmland would alleviate poverty and hunger, facilitate food security and improved nutrition, and encourage the progress of sustainable agriculture. Removing sediment from the reservoir could realize the sustainable management of reservoir water. These policies could also bring a stop to land degradation.

There are 92 reservoirs in Tigray, Ethiopia, about 60% of which have similar problems as the Adizaboy reservoir. If the same methodological approach and technologies could be applied to these reservoirs, farmland areas to be reclaimed using reservoir sediment and irrigation water could be estimated from our water balance calculation. This could be an important basis for proper planning and implementation of sustainable land and watershed management in Ethiopia and beyond.

**Author Contributions:** Conceptualization, K.K., G.G. and F.N.; methodology, K.K. and G.G.; software, K.K.; validation, K.K. and G.G.; formal analysis, K.K. and G.G.; investigation, K.K., G.G. and T.B.; resources, K.K., G.G. and T.B.; data curation, K.K. and G.G.; writing—original draft preparation, K.K.; writing—review and editing, G.G.; visualization, K.K.; supervision, G.G. and T.B.; project administration, F.N.

**Funding:** This research received no external funding.

**Acknowledgments:** The work was conducted as part of the African Watershed Management Project in Ethiopia implemented by the Japan International Research Center for Agricultural Sciences and Mekelle University. The authors sincerely thank Miyuki Iiyama, a research coordinator, and JIRCAS, for advice on the paper writing. Sincere thanks should also go to Mekelle University in Ethiopia for providing a home base and necessary support for this project.

**Conflicts of Interest:** The authors declare no conflict of interest.

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
