# Peer review of "Reservoir Conservation in a Micro-Watershed in Tigray, Ethiopian Highlands"

_sustainability, doi:10.3390/su11072038_

Round 1
Reviewer 1 Report
Dear Authors,
I wish to congratulate for the article, it is very well written, easy to read, I just had 1-2-3 suggestions.
I think that the scientific value is not huge but the importance to the practice is equally important and this is the case here.
Best regards,
Reviewer

Author Response
Point 1: I believe it is tipping and not tripping. (Line No. 145)
Response 1: “Tripping” has been changed to “tipping,” which was incorrectly spelled.
Point 2: no need for dot (Line No. 206)
Response 2: Dot, which was unnecessary, has been deleted.
Point 3: color! (Line No. 217)
Response 3: Red color, which was unnecessary, has been changed to black color.
Point 4: It would be nice to express both in available amount and in mg/kg! (Line No. 220-221)
Response 4: “(25 vs 14.1 cmol(+)/kg (9.8 vs 5.5 g/kg) on bare land)" has been described to express both in available amount and in mg/kg.
Point 5: I think no need for "and" but I let you decide (Line No. 292)
Response 5: “and” has been deleted.

Reviewer 2 Report
The paper presents potential solutions to the problem of sedimentation of retention reservoirs in Ethiopia. The authors have shown that the solution to both soil and water constraints and food security challenges in Ethiopia can be managed by using reservoir sediments for reclaiming farmland while using reservoir water for irrigation.
· More details should be given in the present form, the sentence has no scientific value - „Ecologically and hydrologically related problems of reservoirs in the region have been extensively investigated [3,4,7,8,9,10].”
· Check the values described in the text with those presented in Figure 8.
· Figure 1 - the figure is not easy to read, the use of colours should be considered,
· Do the authors have information on the contamination of bottom sediments of retention reservoirs, e.g. with heavy metals?
· More attention should be given in the Discussion section on the use of bottom sediments by other authors for agricultural purposes, what are the limitations associated with their potential pollution,
· Conclusions need to be corrected, as they should mainly result from the research carried out. Opinions on the potential possibilities and limitations of the suggested solutions should be included in the Discussion section.
Author Response
Point 1: More details should be given in the present form, the sentence has no scientific value - „Ecologically and hydrologically related problems of reservoirs in the region have been extensively investigated [3,4,7,8,9,10].” (Line No. 66)
Response 1: Ecologically and hydrologically related problems of reservoirs in the region have been extensively investigated [3,4,7,8,9,10] regarding the occurrence of cyanobacteria, annual sediment yield of about 19 t/ha/year, and irrigation capacity reduction up to 33% due to water loss through seepage and evaporation, etc.
Point 2: Figure 1 - the figure is not easy to read, the use of colors should be considered (Line No. 90)
Response 2: The line colors have been changed to be easily read in Figure 1.
Point 3: Check the values described in the text with those presented in Figure 8. (Line No. 273)
Response 3: Figure 8 has been modified. Unit of the vertical axis (g/m2) has been consistent with the values described in the text (t/ha).
Point 4: Do the authors have information on the contamination of bottom sediments of retention reservoirs, e.g. with heavy metals? (Line No. 275)
Response 4: No, we do not. I am afraid the we do not have any information on the contamination of bottom sediments.
Point 5: More attention should be given in the Discussion section on the use of bottom sediments by other authors for agricultural purposes, what are the limitations associated with their potential pollution, (Line No. 309)
Response 5: The limitation associated with the potential pollution has been added in the fifth paragraph of the Discussion to give more attention on the use of bottom sediments. “(5) Salt damage might cause the productivity reduction on reclaiming farmland if the salt accumulation occurs. As far as EC values in the reservoir water are concerned, there has been no problem. However, we should conduct the ion chromatography analysis after we use the fertilizer to cultivate the vegetables on reclaiming farmland in the future.”
Point 6: Conclusions need to be corrected, as they should mainly result from the research carried out. Opinions on the potential possibilities and limitations of the suggested solutions should be included in the Discussion section. (Line No. 345)
Response 6: As described in Response 5, the limitation has been added in the Discussion. The last paragraph in the Discussion has been moved to the first paragraph in conclusions.

Reviewer 3 Report
In this manuscript, the authors explored the reservoir conservation in a Micro-watershed in Ethiopian Highlands. This manuscript overall is well written and ready for potential publication. There are some minor comments as follows.
l Figure 4: Numbers 1-7 listed in b and c shall be elaborated either in legend or caption.
l Conclusions: the present form is too brief. The more detailed information of the key findings is needed. Besides, it would be ideal to include some policy recommendations regardingEthiopia’s SDGs, using the key findings of this study.
Author Response
Point 1: Figure 4: Numbers 1-7 listed in b and c shall be elaborated either in legend or caption. (Line No. 212)
Response 1: In figure 4 (b) and (c) numbers 1 to 7 have been presented clearly by caption.
Point 2: Conclusions: the present form is too brief. The more detailed information of the key findings is needed. Besides, it would be ideal to include some policy recommendations regarding Ethiopia’s SDGs, using the key findings of this study. (Line No. 345)
Response 2: Two paragraphs have been added in Conclusions; one is information on the key findings and another some policy recommendations regarding Ethiopia’s SDGs as follows; 1) The results of bathymetric survey using an echo-sounder indicated that the estimated reservoir sediment volume was 6,400 m3 leading to a reclamation of 3.2 hectares by layering 0.2 m sediment. Garlic was produced 7.1 t/ha of bulb on average on reclaiming farmland. There is a potential of producing 2-3 horticultural crops per year. 2) The vegetable cultivation on reclaiming farmland would alleviate poverty and hunger, facilitate food security and improved nutrition, and encourage the progress of the sustainable agriculture. Removing sediments from the reservoir could realize the sustainable management of reservoir water. These policies could also bring a stop of land degradation.
